# Giant sulfur bacteria (Beggiatoaceae) from sediments underlying the Benguela upwelling system host diverse microbiomes

**Beverly E. Flood** [1]*, **Deon C. Louw**[2], **Anja K. Van der Plas** [2], **Jake V. Bailey**[1]

**1** Department of Earth and Environmental Sciences, University of Minnesota, Twin Cities, Minnesota, United States of America, **2** National Marine Information and Research Centre, Swakopmund, Namibia

* beflood@umn.edu

## Abstract

Due to their lithotrophic metabolisms, morphological complexity and conspicuous appearance, members of the Beggiatoaceae have been extensively studied for more than 100 years. These bacteria are known to be primarily sulfur-oxidizing autotrophs that commonly occur in dense mats at redox interfaces. Their large size and the presence of a mucous sheath allows these cells to serve as sites of attachment for communities of other microorganisms. But little is known about their individual niche preferences and attached microbiomes, particularly in marine environments, due to a paucity of cultivars and their prevalence in habitats that are difficult to access and study. Therefore, in this study, we compare Beggiatoaceae strain composition, community composition, and geochemical profiles collected from sulfidic sediments at four marine stations off the coast of Namibia. To elucidate community members that were directly attached and enriched in both filamentous Beggiatoaceae, namely *Ca.* Marithioploca spp. and *Ca.* Maribeggiatoa spp., as well as non-filamentous Beggiatoaceae, *Ca.* Thiomargarita spp., the Beggiatoaceae were pooled by morphotype for community analysis. The Beggiatoaceae samples collected from a highly sulfidic site were enriched in strains of sulfur-oxidizing Campylobacterota, that may promote a more hospitable setting for the Beggiatoaceae, which are known to have a lower tolerance for high sulfide to oxygen ratios. We found just a few host-specific associations with the motile filamentous morphotypes. Conversely, we detected 123 host specific enrichments with non-motile chain forming Beggiatoaceae. Potential metabolisms of the enriched strains include fermentation of host sheath material, syntrophic exchange of $H_2$ and acetate, inorganic sulfur metabolism, and nitrite oxidation. Surprisingly, we did not detect any enrichments of anaerobic ammonium oxidizing bacteria as previously suggested and postulate that less well-studied anaerobic ammonium oxidation pathways may be occurring instead.

## Introduction

Sulfur cycling in marine sediments has been extensively studied and recently reviewed by Jorgensen et al., 2019 [1]. Sulfate, which is highly abundant in marine settings (~28 mM), is the most prevalent driver of the respiration of organic matter under anoxic conditions. The

**Data Availability Statement:** Unaltered fastq files are deposited in the NCBI Sequence Read Archive under Bioproject PRJNA706699. Assembled reads, R commands, and a taxonomy and counts excel

file are available at https://conservancy.umn.edu/handle/11299/219001.

**Funding:** This work was funded by Simons Foundation award #341838 to JVB and the National Science Foundation award #1935351 to JVB and BEF.

**Competing interests:** The authors have declared that no competing interests exist.

product of dissimilatory sulfate reduction, hydrogen sulfide is toxic to the aerobic respiratory pathway. But many bacteria have evolved to couple the oxidation of sulfide with aerobic respiration as well as other electron acceptors. Some of these sulfide-oxidizing bacteria are highly specialized and are dependent upon the oxidation of reduced sulfur ($S_{red}$) species as their primary source of energy. The challenge for these specialists is that $H_2S$ is abiotically oxidized by $O_2$ at fairly efficient rates. As such, they must develop mechanisms and/or relationships that enable them to have access to sources of both $S_{red}$ electron donors and electron acceptors. One such group of sulfur-oxidizing specialists, who have developed some unique adaptations to exploit this lithotrophic niche, is the gammaproteobacterial family, Beggiatoaceae.

The family Beggiatoaceae is composed of 13 named genera of morphologically complex sulfur-oxidizing bacteria [2], some of which are the largest bacteria ever observed [3]. Until recently the Beggiatoaceae were classified as members of the Thiotrichales, which includes other morphologically complex sulfur-oxidizing bacteria such as *Thiothrix* and *Achromatium*. Phylogenic associations based on whole genome comparisons [4, 5] indicate that the Beggiatoaceae are distinctly different from other Thiotrichales, thus reviving the former order Beggiatoales, of which the Beggiatoaceae remain the sole family.

Members of the Beggiatoaceae are chemolithotrophs that often form dense mats in the surface layers of sulfidic sediments, reviewed by [6]. The Beggiatoaceae oxidize inorganic $S_{red}$ species to intracellular stores of cyclooctasulfur ($S_8$) [7, 8] (and polysulfides [9]) that can be further oxidized to $SO_3^{2-}$ and/or $SO_4^{2-}$ [10] or in the absence of extracellular terminal electron acceptors, can be respired to produce $H_2S$ [11–13]. In addition to growth fueled by $S_{red}$ species, some members of the Beggiatoaceae utilize hydrogen as an electron donor under aerobic conditions and for maintenance energy when coupled to the reduction of stored $S_8$ under anoxic conditions [14].

Most strains within the Beggiatoaceae are autotrophs that fix carbon via the Calvin-Benson-Bassham cycle [15] and perhaps in some cases, via the rTCA cycle as well [16, 17]. These autotrophs can also utilize some organic acids as carbon and/or as an energy source, but most have a limited capacity to grow on organic substrates [15, 18–20]. But there are exceptions in which inorganic carbon fixation is not a trait, particularly in freshwater strains which can be either chemolithoheterotrophs [21] or organoheterotrophs [22, 23]. Non-methane methylotrophy has also been reported in some members of the Beggiatoaceae [24, 25].

All of the Beggiatoaceae likely respire aerobically, but many if not most, prefer hypoxic or dysoxic conditions [20, 26–28]. In addition, certain members of the Beggiatoaceae can respire nitrate via the dissimilatory reduction to ammonia (DNRA) or via the denitrification pathway or both [29]. The largest members of the Beggiatoaceae achieve their large size due to a large central vacuole where high concentrations of $NO_3^-$ are stored [3, 30–33] and respired as needed [34]. The dynamics of inorganic nitrogen respiration within the vacuolate marine Beggiatoaceae is complex and not fully understood [16, 35]. Interestingly, nitrogen fixation has been shown to occur within the Beggiatoaceae under nitrogen-limiting conditions [36, 37] but nitrogen fixation has not been detected in vacuolated marine strains.

In general, the Beggiatoaceae can be categorized into two morphological subgroups: filamentous and non-filamentous (coccoidal and chain-forming) strains, Fig 1. The morphology of the Beggiatoaceae is strongly linked to their ecology and responses to environmental conditions. Filamentous clades, i.e. *Beggiatoa* spp., *Candidatus* Maribeggiatoa spp., *Ca*. Isobeggiatoa spp., *Ca*. Parabeggiatoa spp., *Ca*. Allobeggiatoa spp., *Ca*. Halobeggiatoa spp., *Thioploca*, *Ca*. Marithioploca, *Ca*. Marithrix and *Thioflexithrix* spp., glide through the sediments to attractants such as $H_2S$, and away from repellants, such as $O_2$ [10, 26–28, 32] and light [38]. Coccoidal and chain-forming strains, *Ca*. Thiomargarita spp., *Ca*. Thiopilula spp. and *Ca*. Thiophysa spp., are non-motile and must resort to other mechanisms to access substrates and cope with

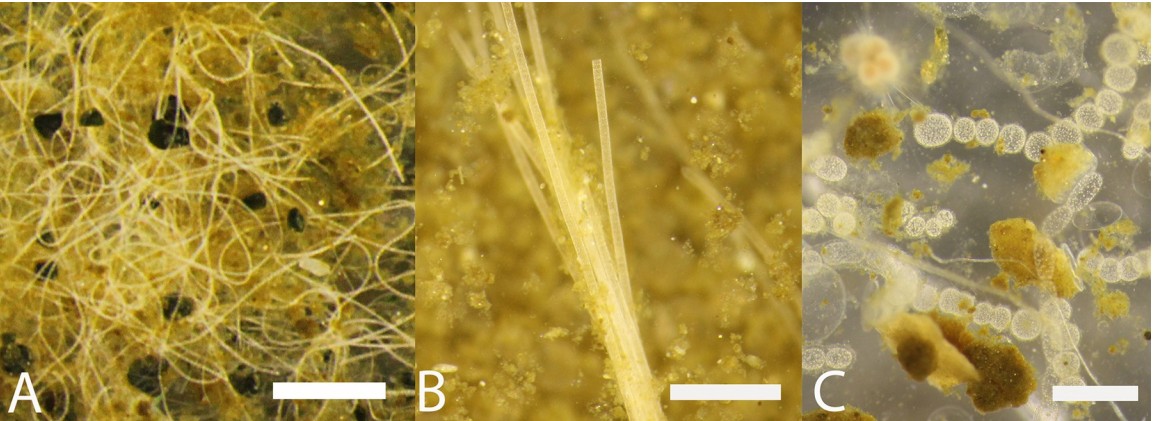

**Fig 1. Light microscopy images of members of the Beggiatoaceae observed onboard the R/V mirabilis, offshore Namibia.** A. Filamentous Beggiatoaceae often morphologically identified as "Beggiatoa", B. Filamentous Beggiatoaceae often morphologically identified as "Thioploca", C. Non-filamentous Beggiatoaceae often morphologically identified as "Thiomargarita". Scale bars in A and B are 150 um, and C in 600 um.

stressful environmental conditions. Unfortunately, there are no coccoidal strains of the Beggiatoaceae in culture and few *in situ* studies [20, 28]. Strains of filamentous Beggiatoaceae cannot be rigorously distinguished from one another based on morphology alone [2], though gross morphological differences are characteristic of certain phylotypes. The same is true for that of the coccoidal chain-forming strains. The Beggiatoaceae can be distinguished from other bacteria based on their large size, general morphology, habitat and the presence of the intracellular sulfur globules.

Shelf sediments under the Benguela Upwelling System off the coast of Namibia, host representatives of most marine genera of the Beggiatoaceae. These sediments are rich in organic matter, primarily derived from phytodetritus resulting in a deposit of diatomaceous ooze. There is little terrigenous input except minor riverine input that is ephemeral and distal [39]. The upwelling of deep ocean waters promotes intensive primary productivity, fueling one of the richest fisheries in the world [40]. The deposition of the decaying pelagic community onto the shelf sediments results in robust microbial respiration and fermentation, which regenerates nutrients to the water column and further stimulates water column productivity [41, 42]. Under these high oxygen demand conditions, the benthic water column can be seasonally hypoxic or even anoxic [43, 44], harming benthic macro-communities [45, 46]. Sulfur-oxidizing bacteria, esp. members of the Beggiatoaceae, are thought to play a crucial role in detoxifying porewater sulfide fluxes into the water column [44] and are a likely food source for poorly characterized benthic biota [47]. Shallow sediments (< 200 m) can contain abundant Beggiatoaceae [3, 44, 48]. *Ca*. Thiomargarita spp. tend to occur in sediments less than ca. 150 m deep where high biological oxygen demand and fluxes of sulfide from the sediments result in bottom water anoxia. *Beggiatoa* and other filamentous strains tend to be found in sediments where bottom waters are hypoxic but not anoxic. Other factors governing ecological zonation of the Beggiatoaceae remain unresolved. Studies of niche partitioning and habitat selection by members of Beggiatoaceae suggest that carbon sources and the ability to store intermediates such as sulfur may play a role [49], as well as the stability of redox gradients [3, 50], differences in growth rates and the absence or presence of nitrate vacuoles [44].

Little is known about the interactions between representatives of the Beggiatoaceae and co-occurring macro- and microbiota. Numerous observations of Beggiatoaceae attaching to marine animals have been made at methane seeps [51, 52] and in marine sediments [53].

Likewise, motile and filamentous Beggiatoaceae are thought to interact with other microorganisms. For example, Beggiatoaceae-dominated mats in marine sediments increase advective flow and produce ammonium and sulfate that are thought to stimulate sulfate reduction and anaerobic ammonium oxidation [54]. Nutrient exchange between sulfate-reducing bacteria and filamentous Beggiatoaceae has also been previously proposed [55, 56]. Some interactions between representatives of the Beggiatoaceae and smaller bacteria may involve physical contact because the large size and production of mucous sheath material can provide a habitat for epibiont bacteria. For example, the sheath of filamentous Beggiatoaceae have been shown to be populated by ammonia oxidizing archaea [57] and annamox bacteria [58]. No molecular studies have been performed on non-filamentous Beggiatoaceae, but bacteria have been previously noted attached to the exteriors of Thiomargarita spp. [59].

In this study, we employed iTag sequencing of the 16S rRNA gene comparing communities associated with pooled filamentous Beggiatoaceae, as well as pooled non-filamentous Beggiatoaceae, and compared these communities with those of their host sediments collected from the continental shelf off of Namibia in southwestern Africa. We hypothesized that intermediates/end products from both sulfur oxidation and nitrate reduction would promote sulfate-reducing and ammonium-oxidizing strains to be significantly associated with both filamentous and non-filamentous Beggiatoaceae, similar to that as previously described in other environments. In addition, we predicted that the comparison of geochemical profiles collected from these sediments with that of the composition of the microbial communities would provide insights into niche partitioning by members of the Beggiatoaceae in these sediments.

## Methods

### Sample acquisition, in situ geochemistry, and sample preservation

Namibian Ministries of Fisheries and Marine Resources approved and supervised sample collection for this study and two scientists who are representatives of the NMFMR are co-authors on the manuscript. An independent field permit was not required. Sediment cores were collected on an April 20–27 2017 expedition of the R/V *Mirabilis* via a MC- 200–4 or MC-400 multi-corer. Samples were collected from four marine stations: Station 23020 (-23.00125, 14.04775; depth 125 m), Station 23002 (-22.999967, 14.3708; depth 36.4 m) Station 20020 (-20.002283, 12.67975; depth 85.8 m), and Station Geochem 2_1 (-24.050,733, 14.250233; depth 118 m). Bulk water column dissolved $O_2$ concentrations were measured via Winkler Titration [60]. Dissolved $O_2$ concentrations in waters above the sediment was measured with a ISO2 Dissolved Oxygen Meter (World Precision Instruments, Sarasota, USA).

Porewater samples were acquired at 1 cm scale horizons from sediment cores collected from Stations 23020, 23002, 20020 but not Geochem2_1. Rhizon CSS samplers (Rhizosphere, Netherlands) filtered the porewater into anoxic Vacutainer® tubes (BD, Franklin Lakes, NJ). Porewater samples, collected to quantify sulfide, were acidified and stored with $ZnCl_2$ (0.5% w/v) to preserve the samples. Porewater samples were stored at 4°C until analyzed. The sediment cores were immediately sectioned. Subsectioned samples were preserved in DNA/RNA Shield™ (Zymo Research Corp., Irvine, CA) and were stored at -20°C during transport. A portion of the top 3 cm of each core and overhead seawater, referred to as "host sediments" was removed and stored at 4°C for collection of filamentous and non-filamentous Beggiatoaceae. Then the host sediments were examined shipboard for the presence of Beggiatoaceae using a Zeiss Primovert inverted microscope. The Beggiatoaceae filaments and chains were rinsed in 0.2 μm filtered seawater and then they were pooled using a micropipette based on morphology, i.e., filamentous samples and non-filamentous samples, until an assemblage of ~100 filaments or coccoidal chains was achieved. Then the Beggiatoaceae samples and the host sediment from

which they were removed were stored separately in DNA/RNA Shield™ at -20˚C. Additionally, upon completion of the collection of Beggiatoaceae samples, the rinse water was preserved in DNA/RNA Shield™ and processed alongside the biological samples as a negative control.

## Porewater analyses

Porewater samples were diluted 100-fold and ran in triplicate for ion chromatography analysis. Major porewater cations ($Li^+$, $Na^+$, $NH_4^+$, $K^+$, and $Mg^{2+}$, and $Ca^{2+}$) were quantified on a Dionex™ ICS 5000+ Capillary HPIC™ system equipped with a CS16-4μm (3 x 250 mm) column and a CERS™ 500 (2 mm) suppressor with an external water regenerant flow and 20 mM methane-sulfonic acid as the eluent. Porewater anions ($F^-$, acetate, formate, $Cl^-$, $NO_2^-$, $Br^-$, $NO_3^-$, $SO_4^{2-}$, $S_2O_3^{2-}$, and $PO_4^-$) were analyzed on the same instrument equipped with a Dionex™ IonPac™ AS19-4μm (2 x 250 mm) column, an AERS™ 500 (2 mm) suppressor, external water regenerant flow and an EG III NaOH eluent generator cartridge. $H_2S$ was measured in triplicate via the methylene blue assay [61] (Product # 2244500; Hach, Loveland, CO).

## iTag analyses

DNA extractions were performed in a LabConco A2 biological safety cabinet using the Zymo-BIOMICS DNA miniprep kit (Zymo, Irvine, CA). The University of Minnesota Genomics Center generated iTag libraries of the V4 hypervariable region as previously described [62]. The V4 primers used were 515F `GTGYCAGCMGCCGCGGTAA` [63] and 806R `GGACTACHVG GGTWTCTAAT` [64]. The Beggiatoaceae samples' rinse waters, and two UMGC water controls were sequenced on one lane of MiSeq for paired-end 2X300 bp reads. Cutadapt v. 2.10 was employed to remove primers and adapters [65]. The paired end reads were assembled using DADA2 v1.16.0 [66] with a maximum expected error rate of 2. phiX was removed prior to error detection, merging of pairs and chimera detection. Taxonomic assignment was performed using the Silva database v. 138 [67]. Bioinformatic and statistical analyses was performed using tools in PhyloSeq 1.32.0 [68]. R package Decontam 1.8.0 [69] identified contaminating amplicon sequence variants (ASVs). Seven ASVs of the genera *Chryseobacterium*, *Lysinibacillus*, *Bradyrhizobium*, *Cutibacterium*, *Stenotrophomonas*, and unclassified members of the clades Xanthomonadaceae and Desulfobulbales were classified as contaminants and were bioinformatically removed from analyses. Post decontamination, the control samples and four samples of low abundance (<15,000 ASVs) were removed prior to further analysis within PhyloSeq. Normalization via variance stabilization transformation (subsampling), as well as detection of ASVs with statistically different abundances, were performed with DESeq2 v. 1.28.0 [70]. The DESeq2 test and parameters for detecting statistically different abundances employed a Wald test with a parametric fit, Cook's distance filtering, and alpha (minimal P-value) set to 0.01. ASVs DESeq2 selected as unclassified at the phylum level were queried (blastN) against the National Center for Biotechnology Information (NCBI) non-redundant nucleotide collection. The top scoring accession was downloaded and queried alongside the unclassified ASVs against the Silva database using the SINA aligner 1.2.11 and were aligned and classified with a rejection below 70% identity. The SINA aligner generated a neighbor joining tree via RAxML using the GTR model and Gamma rate model for likelihoods. The tree was beautified in Figtree v.1.4.4. Additionally, Adobe Illustrator 2020 was used to beautify all images and to generate the modified Venn Diagram.

## Results and discussion

We collected sediment samples from four marine stations- Station 23002 (36.4 m deep), Station 20020 (85 m), Station GeoChem2_1 (118 m) and Station 23020 (125 m), Fig 2. Both $O_2$

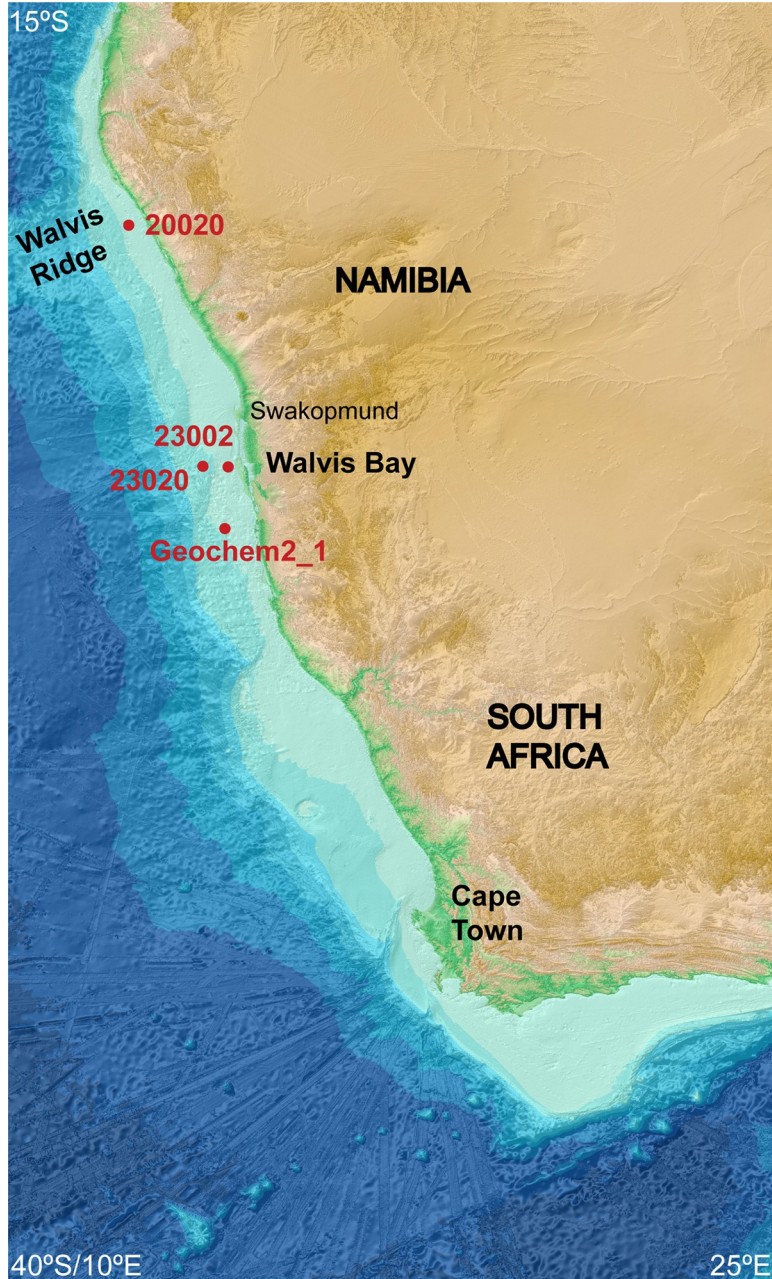

**Fig 2. Map of sampling stations off the coast of Namibia, Africa.** The sampling sites in this study were Marine Station 23002 (-22.999967, 14.3708), depth 36.4 m; Station 23020 (-23.00125, 14.04775), depth 125 m; Station 20020 (-20.002283, 12.67975), depth 85.8 m; and Station Geochem2_1 (-24.050,733, 14.250233), depth 118 m. The baseline map was reproduced from the GEBCO world map 2021, www.gebco.net.

profiles of the water columns (S1 Fig) and microprofiling of the bottom water above the cores indicate that all stations were either hypoxic or anoxic. Stations 23002 and 23020 are the closest stations to one another along the -23 latitude axis but geochemical profiles, Fig 3, and Principal Coordinate Analysis, Fig 4, as well as hierarchical clustering via Euclidean distance matrix (S2 Fig) indicate that the two sampling stations at deeper water depths were more similar to one another, while the other sites were distinctly different. Measures of richness, Chao1 and

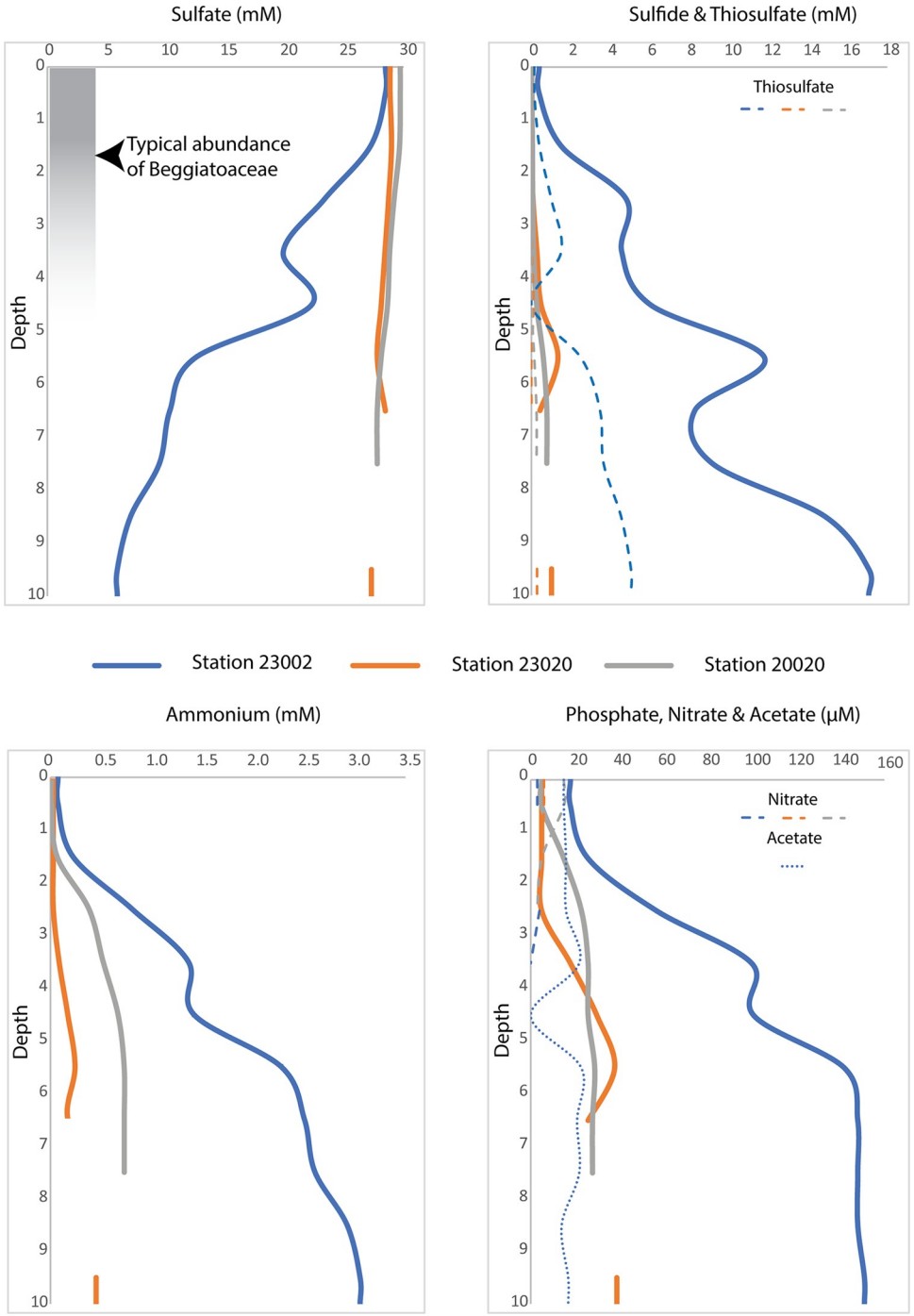

**Fig 3. Geochemical profiles for the top 10 cm of sediment core porewaters with general zonation of Beggiatoaceae.** Porewaters were collected from marine stations 23002 (blue; 1–10 cm), 23020 (orange; 1–7, 9–10 cm) and 20020 (grey; 1–7 cm). Formate (<11 µM) and nitrite (<2 µM) and other data points not shown for analyzed 1cm horizons were below detection limits. Standard deviations were low and representative crossbars were not visible on plots.

Shannon diversity index, indicate that communities from the deepest station, Station 23020, had the greatest diversity and richness, S3 Fig, while communities from the shallowest station, Station 23002 were the least diverse. Despite normalization, differences in DNA library sizes

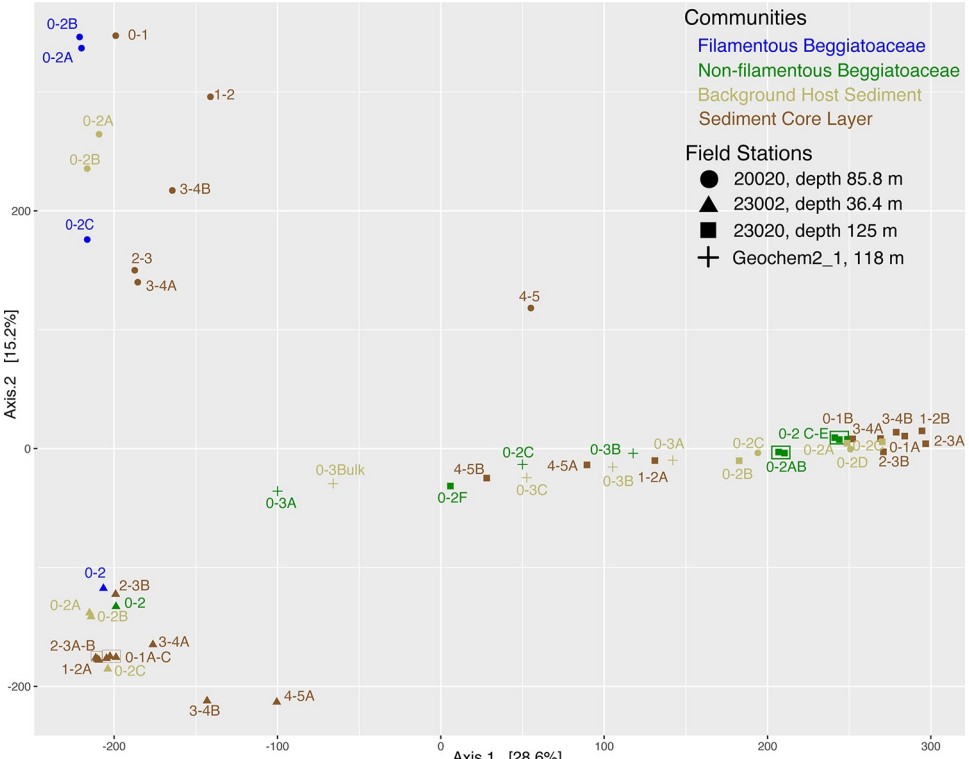

**Fig 4. Principal coordinate analysis of 16S rRNA communities of samples collected from four marine stations: 23002, 20020, 23020 and Geochem2_1 and at depths between 0–5 cm.** The plot was generated using multidimensional scaling with Euclidean distancing.

likely affected many of these analyses whereby some samples were sequenced at a depth that provided substantial characterization of the microbial community, but a few samples were not, S4 Fig.

## Both morphotypes occur at shallow Station 23002

Station 23002 is found within the diatomaceous mud belt [71] but it lies much closer inshore than that of the other mud belt stations and thus likely encounters greater terrigenous input. This is only station where we were able to collect samples of both filamentous and non-filamentous Beggiatoaceae bacteria. But bacteria from both groups of Beggiatoaceae morphotypes were not abundant at this inshore station, thus we were only able to collect one pooled sample of each morphotype from the top 3 cm of the sediment core. The Beggiatoaceae ASVs in the filamentous samples were predominantly the candidate strains *Ca.* Marithioploca, *Ca.* Maribeggiatoa and *Ca.* Parabeggiatoa. These ASVs were also detected in the sediment core with abundances trailing off at or above a depth of 3 cm beneath the sediment/water interface. Other Beggiatoaceae ASVs were also detected in the filamentous sample including *Thioflexithrix* and unclassified Beggiatoaceae ASVs. To the best of our knowledge, this is the first report of a *Thioflexithrix* from a marine environment. The recently discovered type strain, *T. psekupsensis* D3, was isolated from a thermal sulfidic spring [37]. The non-filamentous sample contained a mix of *Ca.* Thiomargarita nelsonii, *T. namibiensis*, *Ca.* Marithioploca, *Ca.* Maribeggiatoa, and an unclassified ASV. ASVs of *Ca.* Thiomargarita were barely detected in the sediment core. But *Thiomargarita's* sister taxa, *Ca.* Thiopilula, was detected throughout the

top 5 cm of sediment core. We were not surprised by the relatively low abundance of Beggiatoaceae ASVs in the core samples and even in the pooled samples. Indeed, low DNA amplification of Beggiatoaceae has been previously observed in these sediments [72]. Many marine strains have introns within their 16S rRNA gene [73] which are believed to be inhibitory to PCR amplification, (Salman-Carvalho *personal comm.* and our own experiences).

Geochemical profiles of the top 10 cm of the sediment core collected from Station 23002 suggest that these sediments are experiencing extensive sulfate reduction near the sediment/water interface Fig 3. Shifts in dominant taxa that may be associated with major metabolic regimes were evident at discreet horizons. Box plots of the most abundant phyla and proteobacterial classes, Fig 5, indicate that the community composition of the top 5 cm of the sediment cores contained a larger fraction of Caldatribacteriota, Cyanobacteria (predominantly diatom chloroplasts but free-living Cyanobacteria cannot be ruled out [74]) and Campilobacterota (formerly known as Epsilonproteobacteria) than other cores. The Verrucomicrobiota and ASVs unclassified at the phylum level composed a smaller fraction of the core community than the other marine stations. Caldatribacteriota is a newly named phylum that includes the former candidate phylum Atribacteria (OP9/JS1), which are thought to be primarily fermenters [75] that may have a syntrophic metabolism with methanogens [76] and that are commonly found in abundance in organic-rich anoxic marine sediments. Fermentation, such as that carried out by representatives of the Caldatribacteriota, is potentially the primary metabolism responsible for the increase in ammonium at depth [77] along with dissimilatory reduction of nitrate [78, 79]. Between 4–5 cm sediment depth, 1 mM of ammonium was consumed concomitantly with thiosulfate and sulfide, generating an increase in sulfate. Around 7 cm of depth, there was a significant drawdown of sulfate. A previous study of nearby sediments indicate that drawdown of sulfate is coupled to AOM [44]. We did not perform iTag sequencing on sediments below 5 cm. But ASVs of ANME-1a, ANME-1b and ANME-2c archaeal anaerobic methylotrophs were present throughout the top 5 cm of the core along with abundant and diverse archaeal methanogens.

One reasonable interpretation of these geochemical profiles is that the horizon at 4–5 cm reflects intense anammox activity coupled to sulfur oxidation and nitrate reduction [58, 80]. However, our iTag data indicated that the anammox bacteria in these sediments, *Ca*. Scalindua spp., were present primarily in the top 3 cm and were almost absent between 4–5 cm of depth. Since anammox bacterial abundance has been shown to be directly correlated to anammox activity [81], it seems likely that other processes were removing $NH_4^+$ at this horizon. However, the iTag community profiles provided little insight into the origin of the drawn down of $NH_4^+$ at this horizon. For an in-depth analysis and discussion, please see S1 Text.

## Only filamentous Beggiatoaceae occur at Station 20020

Marine Station 20020 lies outside of the diatomaceous mud belt. The Beggiatoaceae in the filamentous samples were less diverse than those found at Station 23002 and non-filamentous Beggiatoaceae were barely detectable in these sediments. The dominant Beggiatoaceae ASV was also the most abundant at Station 23002, an ASV of *Ca*. Marithioploca. The second most abundant ASV at Station 20020 was an unclassified member of the Beggiatoaceae. Porewater chemistry from the top 7 cm of the core collected from Station 20020 indicated that the top cm of the core was likely hypoxic while low rates of sulfate reduction, fermentation and DNRA occurred below. There was a notable shift in community composition between the 4–5 cm horizon, where *Ca*. Marithioploca was not detected. Two phyla in the top 4 cm, Chloroflexi and Thermoplasmatota, represented a smaller percent of community abundance in comparison to other stations, Fig 4. The Chloroflexi found at the other field stations were

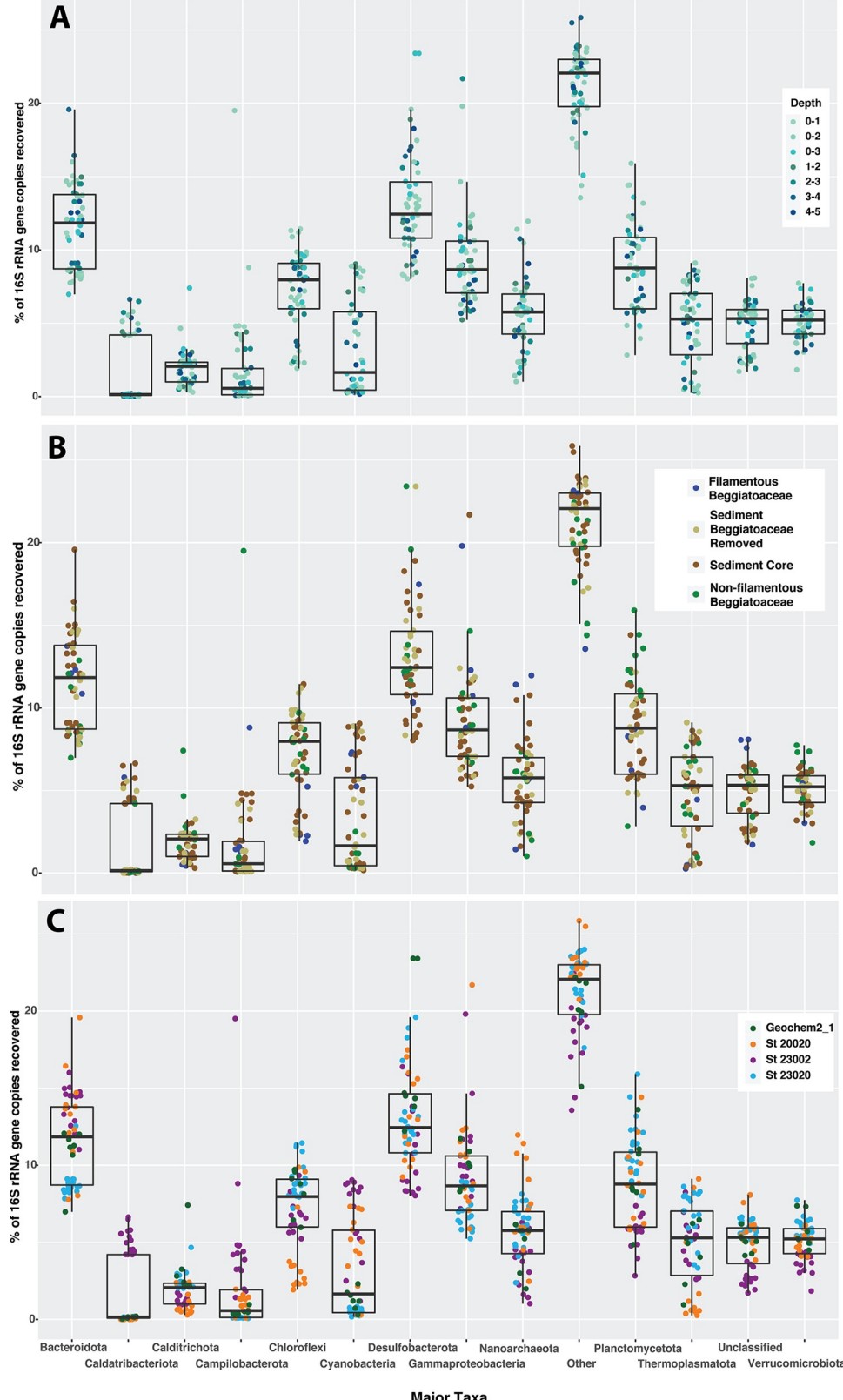

**Fig 5.** Box plots of the phyla of ASVs by percent community composition by A. depth in cm. B. sample type. C. location.

predominantly within the Anaerolineales, which are facultative or strict anaerobes that ferment a broad range of substrates [82]. Thermoplasmatota is a new phylum composed of many previously uncharacterized clades [83]. The most notable difference between the sediments here vs. other stations was the absence of one the most abundant ASVs (ASV #4) in our entire dataset, which was classified as a member of Marine Benthic Group D and DHVEG-1. Representatives of this clade are known to be restricted to anoxic environments [84]. These sediments had an elevated abundance of "cyanobacteria" but again top blastN scores indicate that these ASVs may be plastids of benthic foraminifera, esp. *Virgulinella fragilis*. Perhaps bioturbation generated by vertical migration of filamentous Beggiatoaceae and benthic foraminifera coupled with reduced organic detrital input resulted in the upper 4 cm of sediment being dysoxic, and not anoxic. *Ca*. Marithioploca spp. from the Peruvian margin are thought to be transient in nature occurring in sediments transitioning from hypoxic to anoxic conditions brought about by seasonal and other climatic changes [85, 86].

## Non-filamentous Beggiatoaceae occur in mud belt stations

Initial studies concerning *Ca*. Thiomargarita niche preferences concluded that *Ca*. Thiomargarita abundances were significant when the sediments were euxinic and there was a high flux of sulfide into the water column [48]. A subsequent study monitored four marine stations over a period of three years, measuing both sulfide and methane concentrations as well as the number of *Ca*. Thiomargarita and concluded that there was no correlation between *Ca*. Thiomargarita abundances with porewater and bottom water chemistry [87]. In our study, non-filamentous Beggiatoaceae were very abundant at Marine Station 23020. We were able to collect six replicate samples of ~100 chains each. The two dominant Beggiatoaceae ASVs were *Thiomargarita namibiensis* and *Ca*. T. nelsonii. In much lower abundances were strains of *Ca*. Thiopilula and unclassified Beggiatoaceae.

Marine Station 23020 is in the heart of the diatomaceous mud belt. Total organic carbon can exceed 10 wt.% in these sediments [39]. Our geochemical porewater profiles of sulfate, Fig 2, indicate that Marine Station 23020 was more anoxic in comparison to Marine Station 20020. Yet these organic-rich sediments had low levels of sulfide accumulation and sulfate drawdown, potentially indicating that limited sulfate reduction and sulfide oxidation was occurring in these sediments. However, an alternative interpretation is that the rates of both metabolisms were roughly equal, a scenario has been observed at nearby shelf sediments containing *Ca*. Thiomargarita (23˚46.520 S, 014˚17.960E, 110 m water depth) [88].

Sediment pore water horizons that are characterized by high levels of inorganic phosphate, which are typically found below the sediments containing *Ca*. Thiomargarita spp. [89], are less enriched in phosphate in our samples. It is certainly possible that less inorganic phosphate reaches these sediments than in some other sediments with abundant *Ca*. Thiomargarita spp. Or that the phosphate was sequestered as polyphosphate within the cells at the time of collection. Another possible explanation is that these sediments have undergone methane gas ebullition in the recent past resulting in suspension of sediments and the mixing of porewaters with bottom waters. Periodic suspension of host sediments has been proposed to be the primary mechanism for *Ca*. Thiomargarita to replenish its nitrate stores [3]. Indeed, large methane gas pockets were detected in 2004 in the region of Marine Station 23020 [90]. Alternatively, perhaps these sediments were disrupted by bioturbation but potential bioturbators were not observed and these frequently anoxic sediments are typically depauperate in benthic invertebrates [91].

At Marine Station GeoChem2_1, a sediment core was not collected for geochemical profiling and characterizing community composition 1 cm horizons. Instead, the top 0–3 cm of

surface sediments collected for three pooled non-filamentous Beggiatoaceae samples for comparison with their host sediments and one unaltered "bulk" sample was retained for comparison. These shelly phosphatic sand sediments are well characterized by previous studies (Marine Stations 229 and 12810) [92, 93]. Porewater phosphate concentrations can reach 0.5 mM [94]. *T. namibiensis* and *Ca*. T. nelsonii were roughly equal in concentration in the non-filamentous Beggiatoaceae samples of station GeoChem2_1. We also recovered some *Ca*. Maribeggiatoa and *Ca*. Thiopilula in the non-filamentous Beggiatoaceae samples.

Euclidean clustering of samples from all communities resulted in community ASV compositions of Marine Station 23020 and GeoChem2_1 being more similar to one another than to the other two stations, S2 Fig. In general, at Marine Station 23020 the Bacteroidota, Campilobacterota, Cyanobacteria, and Gammaproteobacteria comprised less of the community than other marine stations. Sediments from this location were in general more diverse than stations where filamentous Beggiatoaceae were present, S3 Fig.

## ASVs uniquely associated with filamentous Beggiatoaceae

We collected three samples of ~100 chains of filamentous bacteria from Marine Station 20020 but only one sample from the shallow Station 23002. Thus, we were able to employ DESeq2, S1 Table, to detect statistically significant associations of ASVs with the filamentous Beggiatoaceae at Station 20020 but not Station 23002. DESeq2 is an R package that was original designed for detecting differential expression between two treatments in RNA sequencing experiments and accounts for differences in sample library sizes. Significant associations were determined through a Wald's Test (p-value > 0.01) adjusted for potential false discoveries (padj). DESeq2 did not identify an ASV in any significant abundance (>40 counts) that was positively correlated with the filamentous Beggiatoaceae samples vs. their host sediments at Station 20020. One of three filamentous Beggiatoaceae samples from Station 20020 had a significantly smaller library size (30,000 vs > 250,000), S4 Fig, which we believe affected the results of the analysis and that there were other likely significant associations to be found. Examples include two strains of the Crenarchaeota *Ca*. Nitrosopumilus which were enriched in the two Station 20020 samples with greater sequencing depth (> 800; >100) vs. their host sediments (< 225; <60). This host-microbe association has been observed with *Ca*. Maribeggiatoa of the hydrothermal system of the Guaymas Basin, Gulf of California [57]. Archaeal aerobic ammonia-oxidizers, such as *Ca*. Nitrosopumilus, can tolerate very low oxygen conditions [95], have high rates of activity [96], and were present in all stations in this study, but were particularly abundant in the top 0–2 cm. But unlike samples from Station 20020, no enrichment was evident in the filamentous Beggiatoaceae sample from Station 23002. Instead, a different aerobic ammonia-oxidizer in low abundance was enriched in this Beggiatoaceae sample, a Nitrosococcaceae SZB85 (49 vs. 0). Members of the Nitrosococcaceae can couple denitrification with sulfide-oxidation under anoxic conditions [97]. Their association with motile Beggiatoaceae would be highly beneficial to both the epibiont and host under hypoxic conditions, but potentially competitive under anaerobic conditions.

A host specific relationship between the filamentous sulfate-reducer *Desulfonema*, and *Ca*. Marithioploca was previously identified in Chilean sediment samples [98] as well as with "Beggiatoa" of the Frasassi Cave System in Italy [56]. Also, significant horizontal gene transfer between the Beggiatoaceae and a strain of *Desulfonema* has been suggested [99]. We found that the filamentous Beggiatoaceae sample from shallow Station 23002 had an ASV classified as *Desulfonema* that was more abundant than that ASV in the host sediments (117 vs. ~5). ASVs of *Desulfonema* were abundant in Beggiatoaceae samples from Station 20020, but not more so than the host sediments. Nor did we detect a potential specific association between

the anammox bacteria, e.g. *Ca*. Scalindua, and Beggiatoaceae bacteria, which were present in the host sediments at both locations.

At Station 23002, the most abundant and enriched ASV in the filamentous Beggiatoaceae sample was the sulfur-oxidizer Campylobacterota, *Sulfurovum*. ASVs of *Sulfurovum* were also very abundant in these sediments, as well as the non-filamentous Beggiatoaceae sample from this location. The genus is usually thought to occur as small rods, but large filamentous strains do exist [100, 101]. Thus, it is possible a filamentous strain of *Sulfurovum* were collected alongside that of the Beggiatoaceae but *Sulfurovum* does not possess intracellular sulfur globules, a distinguishing characteristic of members of the Beggiatoaceae. Other ASVs noticeably-enriched in the Station 23002 filamentous Beggiatoaceae sample were uncharacterized gammaproteobacteria from the orders CH2b56 and eub62A3, but neither were enriched in the Station 20020 samples. In filamentous Beggiatoaceae samples from Station 20020, we identified several bacteria unclassified at the phylum level that may have an association with these Beggiatoaceae including the second most abundant ASV in the sample, as well as a nanoarchaeon ASV. None of the ASVs that appeared to be enriched in samples from Station 20020 were enriched in filamentous Beggiatoaceae from station 23002. We tried to further explore the identity of the second most abundant ASV in the 23002 sample by blasting it against the NCBI non-redundant nucleotide collection, but the closest match had only an 84.52% identity. The most abundant ASV in all three station 20020 filamentous Beggiatoaceae samples was likely a plastid from the benthic foraminifera *Virgulinella fragilis* (100% JN207219). Of the larger two sample DNA libraries the foraminifera plastid exceeded 15,000 ASVs vs 0–4500 in the six host sediment samples. Protists have been found inside the sheath of cold seep *Ca*. Marithioploca spp. from Monterey Bay [102]. And it was postulated that the enclosed sheath environment, where the concentration of hydrogen sulfide is likely lower than ambient conditions, may serve as a refuge for hypoxia tolerant protists like *Virgulinella fragilis*.

## ASVs uniquely associated with non-filamentous Beggiatoaceae

We collected six samples non-filamentous Beggiatoaceae from Station 23020, three samples from GeoChem2_1 and one sample of from the shallow Station 23002. Thus, we used DESeq2 to identify ASVs significantly associated with the non-filamentous Beggiatoaceae from Stations 23020 and GeoChem2_1, S1 Table, but not Station 23002. DESeq2 detected 55 ASVs in Station GeoChem2_1 and 100 ASVs in Station 23020 samples with significance association with host non-filamentous Beggiatoaceae, presented in a modified Venn Diagram, Fig 6. Thirty-two of those ASVs were found to be significantly associated with non-filamentous Beggiatoaceae from both stations. Closer examination of the raw data for the other 91 ASVs indicated that there were other likely significant associations not detected by DESeq2 because of low abundance in one non-filamentous Beggiatoaceae replicate or low-level abundance in host sediment from which the non-filamentous Beggiatoaceae was removed. We found that all but three ASVs from the collective 123 unique ASVs were statistically significantly enriched in the non-filamentous Beggiatoaceae samples between the two stations. We also queried the raw data of shallow Station 23002 for the presence of the 123 enriched ASVs at the other two locations and then compared their abundances verses the host sediment and sediment cores. Thirty-five positive results are highlighted in Fig 6 by larger circles representing ASVs.

The most abundant phyla with significant association with the non-filamentous Beggiatoaceae was the Planctomycetota, the phylum containing anammox bacteria. But none of the 33 Planctomycetota ASVs were of clades for which anammox is known to occur. Instead, 10 ASVs were unclassified at the class level, 11 were from the OM190, 6 from the Phycisphaerales, and 5 were from the Pla4 lineage. One of the unclassified ASVs was also very enriched (300 vs.

## ASVs Associated with Non-filamentous Beggiatoaceae
### p-value < 0.01

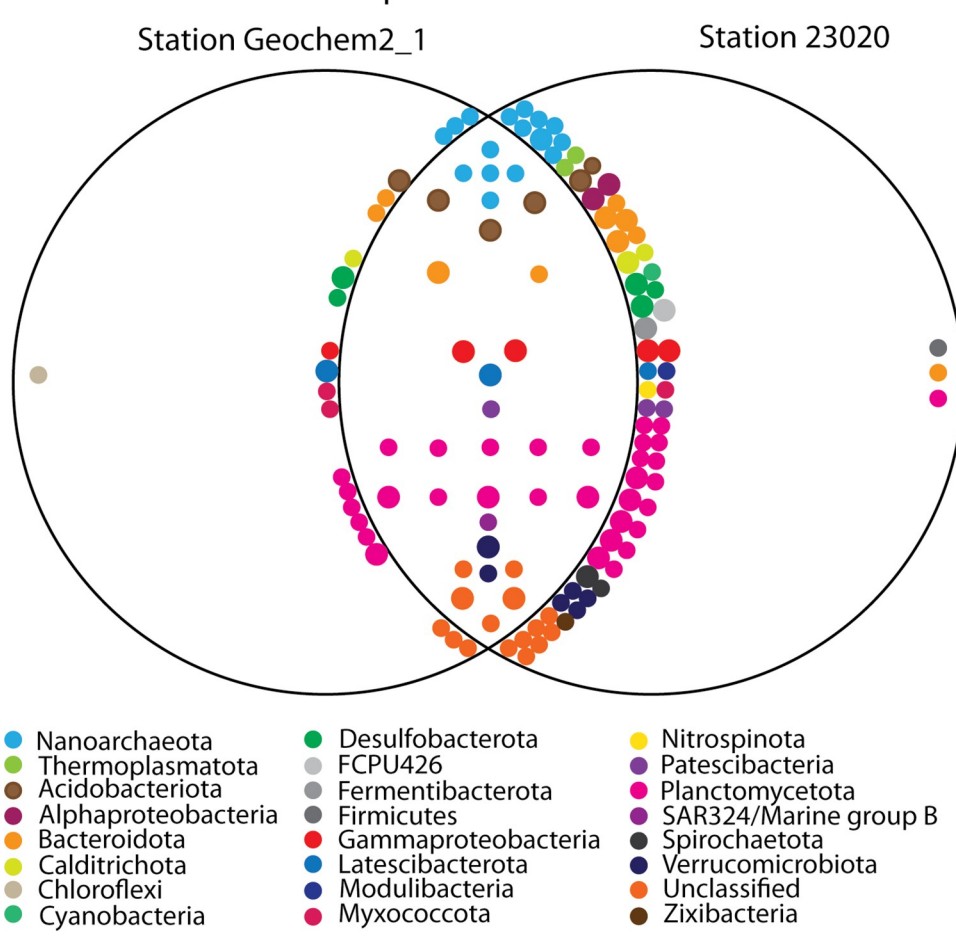

**Fig 6. Modified Venn diagram of 123 ASVs detected by DESeq2 as having a statistically significant association with non-filamentous Beggiatoaceae hosts from Marine Station Geochem2_1 and Marine Station 23020.** ASVs represented in both circles were found to have a significant association with the hosts from both stations. ASVs neighboring the overlapping circles, we determined likely also were significantly enriched the other station but a single low representative inhibited detection via DESeq2. ASVs represented by large circles appeared to also be enriched in the sample from shallow station 23002.

<20) in the sample from the shallower site, Station 23002. One low-abundance ASV found in Station 23020 samples from the Pla3 lineage was not detected in either of the other two station samples. Members of both the OM190 clade and the Phycisphaerales have been found to be attached to macroalgae and are thought to be organoheterotrophs that degrade algal sulfated polysaccharides [103]. Members of the Planctomycetota from the Benguela Upwelling OMZ have been shown to possess a large number of sulfatases [104]. We hypothesize that certain Planctomycetota may attach to *Ca.* Thiomargarita's sheath to degrade sulfated polysaccharides and other sheath constituents.

There were other clades of chemoorganotrophs/fermenters associated with the non-filamentous Beggiatoaceae samples. Members of the Bacteriodota are known to degrade sulfated polysaccharides [105] and have been previously observed attached to the sheath of filamentous *Sulfurovom* [101]. Of the ten ASVs within the Bacteriodota that were found to be associated with the non-filamentous samples, three were within the Bacteroidetes BD2-2 family, two

were within the Flavobacteriaceae, one in the Lentimicrobiaceae, two in the Melioribactera-ceae, and two were within unclassified clades of the Bacteroidales. Other chemoorganotrophs/ fermenters included two ASVs of Kiritimatiellae (Verrucomicrobiota) [106]; three Latescibac-terota [107]; three Calditrichota [108]; three Myxococcota [109]; six Acidobacteria (predomi-nantly subgroups 22 and 23) [110], two Spirochaetota [111] and one ASV each of Lentisphaeria (Verrucomicrobiota) [112], candidate phylum FCPU426 [113], Fermentibacter-ota [114], Ardenticatenales Chloroflexi [115], Gastranaerophilales cyanobacterium (aka. Mel-ainabacteria) [116], unclassified Firmicutes [117], Modulibacteria [118] and Zixibacteria [119]. Most of these clades are poorly studied, but there have been many observations of mor-phologically-distinct members of the Spirochaetota in association with the Beggiatoaceae [120–122]. Many of these clades likely respire nitrogenous species and/or generate $H_2$ and/or low molecular weight organic acids as byproducts of fermentation. *Ca*. Thiomargarita spp. have the capacity to metabolize both acetate and hydrogen [20]. Thus in some cases, chemoor-ganotrophs/fermenters may be beneficial to their Beggiatoaceae host [101]. Additionally, syn-trophic relationships likely exist concerning the exchange of metabolic cofactors such as B-vitamins [123]. Some of these ASVs were also noticeably enriched in the sample from station 23002, e.g., a Bacteroidetes BD2-2 (1400 vs. < 20); and three Subgroup 23 Acidobacteriota (450 vs. 0)(2x 300 vs.<10). Also enriched in the non-filamentous Beggiatoaceae sample from Station 23002 (but not the other two stations) was an ASV of an unclassified Planctomycetota (300 vs. <20); a Maritimimonas flavobacterium (300 vs. 0); a Turicella of Actinobacteriota (300 vs. 0); a Spirochaeta_2 (300 vs. <20); and an Acinetobacter (250 < 0).

In addition to free-living chemoorganotrophs and fermenters, a number of ASVs may depend on the host *Ca*. Thiomargarita spp. for their nutrition either in a syntrophic or preda-tory relationship. Both stations had an ASV of a Patescibacteria (*Ca*. Komeilibacteria family) which have reduced genomes [124] and are thought to be dependent on chemolithoautrophic hosts for intermediates [125]. Nanoarchaea was one of the most abundant phyla in at all of the marine stations, 99.92% of which are within the class Woesearchaeales. Like the Patescibac-teria, the Woesearchaeota have reduced genomes and are thought to be dependent on hosts for metabolic intermediates [126]. Liu et al, 2018 proposed that methanogens would be an ideal host because the Woesearchaeota are capable of acetate fermentation and hydrogen pro-duction which are key to many methanogenesis reactions. But likewise, acetate and hydrogen can also be utilized by members of the Beggiatoaceae. DESeq2 found 15 Woesearchaeota ASVs that were significantly associated with non-filamentous Beggiatoaceae. Yet for such an exchange of nutrients to occur, there would likely need to be direct cell-to-cell contact between the epibiont and the host, which would require the patescibacterium and the Woesearchaeota to have the capacity to penetrate the host's sheath. We have observed considerable variability in the thickness of *Ca*. Thiomargarita spp. sheath with barrel shaped *Ca*. T. nelsonii having the thickest sheaths [2] and thus suggest these relationships may exist in a subset of non-filamen-tous Beggiatoaceae morphotypes. ASVs of these two phyla were not enriched in the sample from Station 23002.

Syntrophic interactions have been demonstrated between sulfate-reducing bacteria and sul-fur-oxidizing bacteria [127–129] and putative sulfate-reducing bacteria have been previously observed on the exteriors of *Thiomargarita* [59]. As anticipated based on these previous obser-vations, putative sulfate reducers were significantly associated with the non-filamentous Beg-giatoaceae of the phyla Desulfobacterota (formerly known as the Deltaproteobacteria) and Thermoplasmatota. Sulfate-reducing Desulfobacterota have been previously identified as hav-ing co-associations with members of the Beggiatoaceae [55, 98]. Some ASVs associated with non-filamentous Beggiatoaceae were in relatively low abundances, i.e. at Station 23020 an ASV of Desulfocaspa and an unclassified Desulfobulbales and at Station GeoChem2_1, an ASV of

Desulfuromusa. Additionally, in the sample from the shallow Station 23002, a Desulfobacterales (300 vs. <20) and a Desulfobulbales (250 vs. 0) appeared to be enriched. Two strains of Desulfosarcinaceae were very abundant in samples from both Station 23020 and Geochem2_1. One strain was also quite abundant in cores (ASV 3) from all four stations but appeared to be enriched in non-filamentous Beggiatoaceae samples from Station 23020. The other strain of Desulfosarcinaceae was only found in non-filamentous Beggiatoaceae samples and not in the sediments from all three stations where the non-filamentous Beggiatoaceae were collected. One replicate from Station 23020 did not have this ASV, and thus it wasn't found by DESeq2 to be statistically significant in its association with non-filamentous Beggiatoaceae at this location. The Desulfosarcinaceae is a metabolically diverse clade but all representatives are sulfate-reducing bacteria. Many strains have been found to be key in syntrophic interactions such as AOM [130], anaerobic ethane oxidation [131] and butyrate-dependent syntrophy [132]. They have also been found to be intracellular symbionts of foraminifera [133] and ciliates [134] in which they provide their fermentative host with amino acids. The potential for sulfate-reduction in ASVs of Thermoplasmatota is suggested by the habitat and unclassified status of the two ASVs that were significant in Station 23020 samples. This class of archaea has been reclassified [83] to include a number of archaeal groups including methanogens (Methanomassiliicoccales), photoheterotrophs *Ca*. Poseidoniales (formerly Marine Group II), and other heterotrophs (Marine Group III and DHVEG-6 [135]. But recently, unclassified members of Thermoplasmatota have been shown to have the genetic potential for sulfate-reduction, hydrogen oxidation and denitrification as well as heterotrophy and fermentation [136].

Surprisingly, putative strains of sulfur oxidizers were significantly associated with non-filamentous Beggiatoaceae. The sulfur-oxidizers were three ASVs of Omnitrophia (Verrucomicrobiota), members of which are thought to oxidize sulfur with iron as an electron acceptor [137], one low abundance ASV of the SAR 324 clade [138], two unclassified Gammaproteobacteria, two Thiotrichaceae Gammaproteobacteria [139] and one highly enriched ASV of an unknown family of the Gammaproteobacteria Incertae Sedis [140], which was also very abundant in the Station 23002 sample (1200 vs. <10). In addition, two ASVs of the Alphaproteobacteria clades Rhizobiales [141] and Rhodobacterales could represent organoheterotrophic sulfur-oxidizers [142]. The ASV of the Rhizobiales was also noticeably enriched in the sample from Station 23002 (200 vs. <10) as well as an unclassified gammaproteobacterium (240 vs. 0). Many organoheterotrophic sulfur-oxidizers, particularly those within the Alphaproteobacteria are very adept at degrading organosulfur compounds, which may be beneficial to co-associated inorganic sulfur oxidizers [143]. The primary sulfur-oxidizers that were enriched in the shallower site, Station 23002, were from the phylum Campylobacterota. Indeed, the most abundant and highly enriched ASV in the non-filamentous Beggiatoaceae sample was a strain of the Campylobacterota *Sulfurimonas* [144] (>16,000 vs. less than 500 in sediment samples). The *Sulfurovum* ASV that was enriched in the filamentous Beggiatoaceae sample from shallow Station 23002 was also abundant but not necessarily enriched in the non-filamentous Beggiatoaceae sample from the same station.

Both the filamentous and non-filamentous Beggiatoaceae were strongly associated with sulfur-oxidizing Campylobacterota at the shallow Station 23002, but not at other stations where the Beggiatoaceae were more abundant. Several studies have proposed a difference in niche preferences between sulfur-oxidizers within the Campylobacterota vs. the Gammaproteobacteria, including members of the Beggiatoaceae. In general, Campylobacterota dominate in sediments with higher concentrations of sulfide potentially due to greater sensitivity to $O_2$ [50, 145–148]. Often these sulfide-rich sediments contain the intermediates of sulfur respiration and disproportionation such as $S^0$. Some Campylobacterota, can incompletely oxidize sulfide to $S^0$ which is then excreted [149, 150] while others, such as strains of *Sulfurimonas*/

*Sulfurovum* are thought to be able to oxidize extracellular $S^0$, including cyclooctasulfur ($S_8$), a trait unique to the Campylobacterota [151, 152]. Some strains of Campylobacterota from hydrothermal vent systems reduce extracellular $S_8$ and/or other S intermediates when $H_2$ is available [153–155], much like respiration of internal sulfur globules by members of the Beggiatoaceae. Whether this occurs in other settings is unknown. One Campylobacterota sulfur-reducer at hydrothermal vents is *Thiofractor thiocaminus*, the sole isolate from this genus [154]. An ASV of *Thiofractor* was the second most abundant ASV in the non-filamentous Beggiatoaceae sample from Station 23002 (8,000 vs. >100 in sediment samples). Whether these Campylobacterota are functioning as sulfur-oxidizers or sulfate reducers in this setting remains to be determined. We propose that attached sulfur-oxidizing Campylobacterota may be key to Beggiatoaceae bacteria inhabiting these highly sulfidic sediments by reducing the concentration of sulfide in the diffusion boundary layer surrounding the Beggiatoaceae cells [156]. Hydrogen sulfide can be deleterious to cells when one electron is oxidized producing certain sulfur intermediates [157, 158]. While both clades must have a variety of protective responses, these Campylobacterota may have evolved more efficient/effective adaptations to maintain cellular homeostasis to include producing extracellular products of the oxidation of sulfide (vs. the more metabolically expensive mechanism of funneling $S_{red}$ into intracellular sulfur globules). The production of extracellular sulfur intermediates by the Campylobacterota may also be coupled to the metabolism of attached strains that can disproportionate sulfur, such as members of the Desulfobacterales [159, 160] creating microenvironments of robust sulfur cycling.

We anticipated that a number of enriched ASVs would be ammonium and nitrite oxidizing bacteria, because inorganic nitrogen cycling could help sustain viability between occasional exposure to $O_2$ [161]. Instead, many of the ASVs associated with the non-filamentous Beggiatoaceae were more likely to be denitrifiers and ammonifiers. If these bacteria respire nitrate, they would be competing with their Beggiatoaceae hosts for that electron acceptor. As in the case of the filamentous Beggiatoaceae, anammox bacteria were present in the host sediments containing these Beggiatoaceae but they were not enriched in the Beggiatoaceae samples. These sediments also contained ASVs from an uncharacterized clade of Nitrospirota. Members of this phylum are known to carry out complete aerobic nitrification of ammonia to nitrate [162, 163], but these ASVs were also not enriched in the non-filamentous Beggiatoaceae samples. Neither were members of the Nitrosococcales that were in low abundances in these sediments. DESeq2 found only one statistically significant association between a potential microoxic nitrifier of nitrite to nitrate within an unclassified clade distal to known nitrite nitrifiers within the Nitrospinota [164, 165]. This ASV was of low abundance but was clearly enriched in samples from Station 23020, and to a lesser extent, samples from Station GeoChem2_1.

We were unable to find any indication of enrichment of autotrophic ammonium or nitrite oxidizers in the sample from the shallow site, Station 23002. There was some indication of enrichment of clades known for heterotrophic nitrification, like members of the Pseudomonadales and Comamonadaceae, which were not detected in the host sediment. This metabolic capability has been poorly studied outside of wastewater treatment, e.g., [166, 167], and the clades capable of heterotrophic nitrification, particularly in marine settings, have not been well explored. In some cases, they are mixotrophs that possess ammonia monooxygenases and hydroxylamine oxidoreductases homologous to those of autotrophic nitrifiers, thus they likely derive energy from them [168]. But in other cases, the enzymes appear to be of a different origin [167, 169–172]. These heterotrophic nitrifiers are also often capable of aerobic denitrification and are sometimes sulfur-oxidizing bacteria [173]. This metabolism is thought to conserve $O_2$ for respiration under low $O_2$ conditions by reducing the electron flow to $O_2$ [174]. Some aerobic ammonia-oxidizers and nitrifiers can couple these processes to

denitrification [167, 175] under anaerobic conditions, which sometimes results in the release of NO [176, 177] and $N_2O$ [178, 179] or they denitrify using other electron donors such as $H_2$ [180].

Another potential metabolism that would involve cycling of inorganic nitrogenous compounds anaerobically is sulfammox, (Eq 1) [181, 182].

$$8NH_4^+ + 3SO_4^{2-} \rightarrow 4N_2 + 3HS^- + 12H_2O + 5H^+ \; \Delta G_0' = -17.6 \; KJ \; mol^{-1} \; (pH \; 7) \quad Eq1$$

Like heterotrophic nitrification, this process has primarily been studied in wastewater treatment [183] but has recently been detected in marine settings [80, 184]. In many natural settings, sulfammox is not thermodynamically favorable at ambient conditions, often requiring either high ammonium or the removal to the intermediate nitrite. Autotrophic sulfide-oxidizers capable of denitrification have long been implemented in facilitating sulfammox by both removing nitrite but also reduced sulfur species, e.g., [185–189]. Currently, the clade(s) responsible for sulfammox are unknown, but traditional anammox bacteria and other Planctomycetes clades [186–190] including the Phycisphaeraceae [184], as well as members of the Verrucomicrobiales [187–189] have been suggested to play a role in the process. One isolated strain of *Bacillus* sp. was thought to perform sulfammox [191] and recently, a Betaproteobacterium, *Ralstonia* sp., has been isolated that performs this reaction, but this *Ralstonia* sp. prefers thiosulfate over sulfate as an electron acceptor [192]. Currently, the metabolic pathways that support these sulfammox processes are undetermined, but in many cases, nitrite is commonly detected as an intermediate, e.g. [192]. The production of nitrite along with the production of reduced sulfur species would be highly beneficial to members of the Beggiatoaceae. Furthermore, Beggiatoaceae performing DNRA and sulfide-oxidation would be beneficial hosts for sulfammox bacteria that require both sulfate or thiosulfate and ammonium.

Lastly, there were a total of 16 bacterial ASVs that were unclassified at the phylum level, but which were detected by DESeq2 as having a significant association with non-filamentous Beggiatoaceae samples. We attempted to determine their last known common ancestor, S5 Fig. Six of these ASVs were too distant from sequences within the Silva database to infer the ancestral origin. The Acidobacteria appear to be the last common ancestor of five ASVs and the Planctomycetes of two ASVs. Three other ASVs had the common ancestral phyla of the Spirochaetota, Bdellovibrionata, and the Myxococcota. Five ASVs were statically significant in samples from both Station 23020 and Geochem2_1 and one of which was highly enriched in the sample from Station 23002 (1300 vs. <10). The sample from Station 23002 also had an unclassified bacterial ASV that was one of the most abundant ASVs in the sample and was highly enriched comparison to the host sediments (2700 vs. < 100). We postulate that many of these ASVs may represent strains that have evolved to be specifically host-adapted to members of the non-filamentous Beggiatoaceae as either syntrophs or parasites.

## Conclusion

The objectives of this study were two-fold: 1) to learn more about the environmental parameters of the ecotypes of motile (filamentous) vs. non-motile (non-filamentous) members of the Beggiatoaceae; 2) to determine if there may be select microbes preferentially attached or associated with to these of the Beggiatoaceae. Our sediment core pore water geochemistry profiles revealed that both filamentous and non-filamentous Beggiatoaceae can be found under a broad range of oxygen and sulfide concentrations. Under high concentrations of sulfide we observed very abundant host associated members of the Campylobacterota, which are known to prefer regimes with high sulfide to oxygen ratios. We hypothesize that they may reduce the concentration of sulfide around the host Beggiatoaceae promoting an environmental setting

that is more amenable to the Beggiatoaceae that are less tolerant of extremely sulfidic conditions. In these sulfide-rich sediments, we also found enriched ASVs of the *Desulfonema* and foraminifera in the filamentous Beggiatoaceae sample, as previously reported to be associated with filamentous Beggiatoaceae at other environments. These associations were not found in the deeper water settings, where sulfide concentrations are lower in the sediments. However, in deeper settings, another previously reported Beggiatoaceae host association was found, that of the aerobic ammonium oxidizer *Ca.* Nitrosopumilus. In general, we found fewer specific associations between the filamentous Beggiatoaceae and their epibiont community than we found with the non-filamentous Beggiatoaceae, even after taking into account DNA library size biases. But previously unreported associations appear to exist, including those with strains that are unclassifiable at the phylum level.

Outside of the sulfidic shallow sediments, the non-filamentous Beggiatoaceae were predominately found in the heart of the mud belt sediments, which are prone to gas ebullition. We found 123 ASVs to be enriched in Beggiatoaceae from these sediments many of which are likely capable of syntrophic metabolisms supplying their hosts with organic acids and hydrogen. Others may compete with their hosts for nutrients such as nitrate and sulfide or have unknown metabolic potential. We expected to find an association between anammox bacteria and non-filamentous Beggiatoaceae but found no such relationship. Instead, we found a nitrite oxidizer enriched in the host sample relative to the sediment, but in low abundance. Since inorganic nitrogen is key to respiration in these non-motile strains and both energy and carbon sources are likely provided by epibionts, we propose that other less well-studied metabolic processes such as heterotrophic nitrification and sulfammox may be supplying these Beggiatoaceae with nitrate and/or nitrite.

A significant limitation of our study was that the Beggiatoaceae were pooled based on morphology thus combining different genera and species, so exact host association remains elusive. Additionally, phylogenetic identification of taxa using amplicon sequencing is not a perfect predictor of metabolic potential, although many of the ASVs reported on here represent clades with distinctive metabolic characteristics. Whole genome amplification of individual Beggiatoaceae chains and filaments for metagenomic analyses will further elucidate these relationships. Niche preference by individual species could also be improved by analyzing the host sediment particle size and composition, including the percent of organic matter, as well as collecting additional porewater chemistry including other potential microbial substrates such as methane. While the results of this study have left us with more questions than answers, they reveal that the Beggiatoaceae are more than bacteria: they are hosts of their own microbiomes.

## Supporting information

**S1 Fig. Water column dissolved oxygen profiles off Namibia collected as part of the regional graduate networks in oceanography survey April 2017.** A. Along the -20 latitude, B. Along the -23 latitude, C. At Marine Station Geochem2_1 April 2017.
(TIF)

**S2 Fig. Hierarchical cluster analysis via euclidean distance of the 16S rRNA community composition of samples collected at four marine stations: 23002, 20020, 23020 and Geochem2_1 and various depths between 0–5 cm.**
(TIF)

**S3 Fig. Mean species diversity via the Chao1 (sampling depth) and Shannon (evenness of ASVs) indices of the 16S rRNA communities of samples collected from four marine**

stations: 23002, 20020, 23020 and Geochem2_1 and at various depths between 0–5 cm.
(TIF)

**S4 Fig. Rarefaction plots of 16S rRNA communities of samples collected from four marine stations: 23002, 20020, 23020 and Geochem2_1 and at various depths between 0–5 cm demonstrate that some samples were of lower sequencing depth than most samples.** The plots demonstrate that some samples had a small library size which failed to adequate capture the entire community, which likely resulted in under detection via DESeq2 of some significant associations between host Beggiatoaceae and their attached epibionts.
(TIF)

**S5 Fig. Neighbor joining 16S rRNA gene phylogenic distance tree of ASVs with statistically significant association with non-filamentous Beggiatoaceae along with top NCBI blastn match.** The 16S rRNA gene sequences were classified against the Silva database using the SINA aligner 1.2.1.1 with a 70% identity cutoff. Red stars indicated that the ASV was statistically significant for both stations 23020 and Geochem2_1 samples. Blue stars indicate that the ASV was likely associated with the non-filamentous LSB sample from Marine Station 23002.
(TIF)

**S1 Text. Discussion of geochemical profiles of Marine Station 23002 sediments collected April 2017.**
(DOCX)

**S1 Table. DeSeQ2 identified significant ASVs in Beggiatoaceae samples vs. their host sediments for Marine Station 20020, Marine Station 23020 and Marine Station GeoChem2_1.**
(XLSX)

## Acknowledgments

We would like to thank contributions made by Chibo Chikwililwa, Kurt Hanselmann, Bronwen Currie, the crew of the R/V Mirabilis, the staff and students of the Regional Graduate Networks in Oceanography (RGNO) Discovery Camp, the University of Namibia's Sam Nujoma Marine and Coastal Resources Research Centre (SANUMARC), the National Marine Information and Research Center (NatMIRC) of the MFMR, the Scientific Committee for Oceanographic Research (SCOR), the Agouron Institute and the Simons Foundation who fund the RGNO Discovery Camps, which made this field work possible. And we thank the skilled staff of the University of Minnesota's Genomics Center. Lastly, we thank the reviewers and editor for their thoughtful insight.

## Author Contributions

**Conceptualization:** Beverly E. Flood, Jake V. Bailey.

**Data curation:** Beverly E. Flood.

**Formal analysis:** Beverly E. Flood, Deon C. Louw, Anja K. Van der Plas.

**Funding acquisition:** Beverly E. Flood, Jake V. Bailey.

**Investigation:** Beverly E. Flood.

**Methodology:** Beverly E. Flood.

**Project administration:** Jake V. Bailey.

**Visualization:** Beverly E. Flood, Deon C. Louw, Anja K. Van der Plas, Jake V. Bailey.

**Writing – original draft:** Beverly E. Flood.

**Writing – review & editing:** Beverly E. Flood, Jake V. Bailey.

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
