## [Decision Letter · Decision Letter 0]

8 Jul 2021

PONE-D-21-15605

Giant sulfur bacteria (Beggiatoaceae) from sediments underlying the Benguela Upwelling System host diverse microbiomes

PLOS ONE

Dear Dr. Flood,

Thank you for submitting your manuscript to PLOS ONE. After careful consideration, we feel that it has merit but does not fully meet PLOS ONE’s publication criteria as it currently stands. Therefore, we invite you to submit a revised version of the manuscript that addresses the points raised during the review process.

Both reviewers and I think the paper reports new and interesting findings about the Beggitoa microbiome. The reviewers  indicated that the manuscript could be suitable for publication, provided their comments are responded to adequately. In addition to theirs, I suggest a minor editorial comment: on line 31, I think it would be better to say that they were "pooled by morphotype for community analysis" or something similar; the goal of the iTag sequencing is more important than the approach.

We look forward to receiving your revised manuscript.

Kind regards,

John M. Senko

Academic Editor

PLOS ONE

Journal Requirements:

Reviewers' comments:

Reviewer's Responses to Questions

**Comments to the Author**

1. Is the manuscript technically sound, and do the data support the conclusions?

Reviewer #1: Yes

Reviewer #2: Yes

2. Has the statistical analysis been performed appropriately and rigorously? 

Reviewer #1: Yes

Reviewer #2: Yes

3. Have the authors made all data underlying the findings in their manuscript fully available?

Reviewer #1: Yes

Reviewer #2: Yes

4. Is the manuscript presented in an intelligible fashion and written in standard English?

Reviewer #1: Yes

Reviewer #2: Yes

5. Review Comments to the Author

Reviewer #1: The paper “Giant sulfur bacteria (Beggiatoaceae) from sediments underlying the Benguela Upwelling System host diverse microbiomes” authored by Beverly Flood, Deon C. Louw, Anja K. Van der Plas and Jake V. Bailey is devoted to ecological study of microbial communities on the coast of Namibia, where abundant aggregation by bacteria from the family Beggiatoaceae occurs. The authors pay special attention to the composition of communities associated with mucous formations around sulfur bacteria from the family Beggiatoaceae. The authors investigate the geochemical characteristics of water at the sampling sites. To determine the taxonomic composition, the authors sequenced the V4 region of 16S rRNA from the samples.

The authors make a conclusion about metabolic potential of organisms associated with Beggiatoaceae on the basis of the taxonomic composition. These miroorganisms interact with each other through syntrophic exchange of H 2 and acetate produced as the result of Beggiatoaceae sheath fermentation, as well as products of sulfur and nitrogen compounds metabolism.

It is interesting that there are no anaerobic ammonium-oxidizing bacteria (annamox) among Beggiatoaceae-associated microorganisms. But authors suppose that anaerobic/miroaerobic lithotrophic ammonium oxidation could be coupled with either iron (ferromox), sulfate (sulfammox) or thiosulfate (sammox) respiration and/or heterotrophic nitrification as well as denitrification could be occurring.

The work was carried out at a high scientific and methodological level. A large amount of data was obtained, and the authors managed to analyze and structure these date. All results seem to be good and valid. There are some minor comments and corrections on the article:

L. 88: i.e. should not be written in italic. Commas after Beggiatoa spp. and Ca. Isobeggiatoa spp. are needed

Please, delete comments from S1 Text

L. 266: ‘Thioflexithrix’, not ‘Thioflexothrix’. It is interesting that ASVs from marine sediment can be classified as Thioflexithrix, because originally described Thioflexithrix psekupsensis was isolated from freshwater habitat. it might be worth discussing this fact.

L. 485: it may be worth mentioning that the opportunity of Beggiatoaceae assotiation with Spirochaeta was already shown (Grabovich et al., 2021 DOI: 10.1016/j.biosystems.2020.104322)

L. 521: Thiomargarita should be written in italic

Ref 25. Leptomitoformis should be written with a lowercase letter

Reviewer #2: This manuscript, exploring the diversity of Beggiotoa and their symbionts in sediments off the coast of Namibia, was very interesting and pleasurable to read. The findings were well-presented and supported by the data. I have only a few minor comments, mostly concerning grammatical corrections which I feel will improve the clarity of the text.

line 91: "way" should be "away" (I believe)

line 98: remove "But"

line 122: remove "But"

line 132: add hyphen to make it "Beggiatoaceae-dominated"

line 133: remove hyphen to make it "sulfate reduction"

lines 272-274: Is "introns" the correct term here? I was not aware that introns were found in prokaryotic genomes. Also, "which is" should be changed to "which are," if you are referring to multiple introns.

lines 438-450: "Significantly associated" should be further defined here. How does DESeq2 make this determination (I assume with statistical analyses but if so, which ones).

lines 525-527: This sentence is somewhat confusing- consider rephrasing.

line 534: remove "But"

line 535: change "thus" to "and thus"

line 555: "one ASV" but then two orders are named. Does this mean one ASV from each order?

line 706: replace comma after "bacteria" with a colon

6. PLOS authors have the option to publish the peer review history of their article (what does this mean?). If published, this will include your full peer review and any attached files.

Reviewer #1: No

Reviewer #2: **Yes: **Lauren M Seyler

---

## [Author Response · Author response to Decision Letter 0]

15 Sep 2021

Response to Editor

I suggest a minor editorial comment: on line 31, I think it would be better to say that they were "pooled by morphotype for community analysis" or something similar; the goal of the iTag sequencing is more important than the approach. 

Corrected to: To elucidate community members that were directly attached and enriched in both filamentous Beggiatoaceae, namely Ca. Marithioploca spp. and Ca. Maribeggiatoa spp., as well as non-filamentous Beggiatoaceae, Ca. Thiomargarita spp., the Beggiatoaceae were pooled by morphotype for community analysis.

Response to Reviewer #1

L. 88: i.e. should not be written in italic. Commas after Beggiatoa spp. and Ca. Isobeggiatoa spp. are needed. Corrected.

Please, delete comments from S1 Text. Corrected.

L. 266: ‘Thioflexithrix’, not ‘Thioflexothrix’. It is interesting that ASVs from marine sediment can be classified as Thioflexithrix, because originally described Thioflexithrix psekupsensis was isolated from freshwater habitat. it might be worth discussing this fact.

Added: To the best of our knowledge, this is the first report of a Thioflexithrix from a marine environment. The recently discovered type strain, T. psekupsensis D3, was isolated from a thermal sulfidic spring (37).

L. 485: it may be worth mentioning that the opportunity of Beggiatoaceae association with Spirochaeta was already shown (Grabovich et al., 2021 DOI: 10.1016/j.biosystems.2020.104322)

Thanks for catching this! We researched this further and found that there have been observations of such associations going back to 1910! Unfortunately, the 1910 & 1912 manuscripts are not available; but, they were referenced in the Dyar 1947 manuscript (referenced below: Dyar, 1947; Blakemore & Canale-Parola, 1973; and Grabovich et al., 2021). 

Added: “Most of these clades are poorly studied, but there have been many observations of morphologically-distinct members of the Spirochaetota in association with the Beggiatoaceae (120-122). Many of these clades likely …”. (changed to “clades” in the next sentence because “strains” might infer just the Spirochetes vs. all of the clades discussed above).

L. 521: Thiomargarita should be written in italic. Corrected.

Ref 25. Leptomitoformis should be written with a lowercase letter. Corrected. 

Response to Reviewer #2.

line 91: "way" should be "away" (I believe). Corrected to “away”.

line 98: remove "But". Corrected.

line 122: remove "But" Corrected.

line 132: add hyphen to make it "Beggiatoaceae-dominated" line 133: remove hyphen to make it "sulfate reduction" Corrected.

lines 272-274: Is "introns" the correct term here? I was not aware that introns were found in prokaryote genomes. Also, "which is" should be changed to "which are," if you are referring to multiple introns. Corrected to “which are”.

Yes, “intron” is the correct term. Although rare in prokaryotes, introns do occur, particularly in 16S rRNA genes but also occasionally in functional genes. Introns contributed to missing some candidate phyla by classic 16S rRNA gene sequencing. Genomes of marine strains of the Beggiatoaceae often contain both Group I and Group II introns (below references: Salman et al., 2012, Flood, et al., 2016).

lines 438-450: "Significantly associated" should be further defined here. How does DESeq2 make this determination (I assume with statistical analyses but if so, which ones). 

Added the following. “DESeq2 is an R package that was original designed for detecting differential expression between two treatments in RNA sequencing experiments and accounts for differences in sample library sizes. Significant associations were determined through a Wald’s Test (p-value > 0.01) adjusted for potential false discoveries (padj).”

lines 525-527: This sentence is somewhat confusing- consider rephrasing.

Corrected to: “Some ASVs associated with non-filamentous Beggiatoaceae were in relatively low abundances, i.e. at Station 23020 an ASV of Desulfocaspa and an unclassified Desulfobulbales and at Station GeoChem2_1, an ASV of Desulfuromusa. 

line 534: remove "But" Corrected.

line 535: change "thus" to "and thus" Corrected.

line 555: "one ASV" but then two orders are named. Does this mean one ASV from each order? 

Corrected to: In addition, two ASVs of the Alphaproteobacteria clades Rhizobiales and Rhodobacterales could represent organoheterotrophic sulfur-oxidizers. 

line 706: replace comma after "bacteria" with a colon. Corrected.

References

Blakemore, RP, and E Canale-Parola. "Morphological and Ecological Characteristics of Spirochaeta Plicatilis." Archiv für Mikrobiologie 89, no. 4 (1973): 273-89. https://dx.doi.org/https://doi-org.ezp2.lib.umn.edu/10.1007/BF00408895.

Dyar, MT. "Isolation and Cytological Study of a Free-Living Spirochete." Journal of Bacteriology 54, no. 4 (1947): 483-93. https://dx.doi.org/https://doi.org/10.1128/jb.54.4.483-493.1947.

Flood, Beverly E., Palmer Fliss, Daniel Seth Jones, Gregory J. Dick, Sunit Jain, Anne-Kristin Kaster, Matthias Winkel, Marc Mußmann, and Jake Bailey. "Single-Cell (Meta-)Genomics of a Dimorphic Candidatus Thiomargarita Nelsonii Reveals Genomic Plasticity." Original Research, Frontiers in Microbiology 7 (2016-May-3 2016): 603. https://dx.doi.org/10.3389/fmicb.2016.00603.

Grabovich, M. Y., M. V. Gureeva, and G. A. Dubinina. "The Role of the "Thiodendron" Consortium in Postulating the Karyomastigont Chimaera of the Endosymbiosis Theory by Lynn Margulis." Biosystems 200 (Feb 2021): 104322. https://dx.doi.org/10.1016/j.biosystems.2020.104322.

Salman, Verena, Rudolf Amann, David A. Shub, and Heide N. Schulz-Vogt. "Multiple Self-Splicing Introns in the 16s Rrna Genes of Giant Sulfur Bacteria." Proceedings of the National Academy of Sciences 109, no. 11 (Mar 13 2012): 4203–08. https://dx.doi.org/10.1073/pnas.1120192109.

---

## [Editor Report · Decision Letter 1]

21 Sep 2021

Giant sulfur bacteria (Beggiatoaceae) from sediments underlying the Benguela Upwelling System host diverse microbiomes

PONE-D-21-15605R1

Dear Dr. Flood,

We’re pleased to inform you that your manuscript has been judged scientifically suitable for publication and will be formally accepted for publication once it meets all outstanding technical requirements.

Kind regards,

John M. Senko

Academic Editor

PLOS ONE
---

## [Editor Report · Acceptance letter]

10 Nov 2021

PONE-D-21-15605R1 

Giant sulfur bacteria (Beggiatoaceae) from sediments underlying the Benguela Upwelling System host diverse microbiomes 

Dear Dr. Flood:

I'm pleased to inform you that your manuscript has been deemed suitable for publication in PLOS ONE. Congratulations! Your manuscript is now with our production department. 

Kind regards, 

on behalf of

Dr. John M. Senko 

Academic Editor

PLOS ONE